# The Emergence of Senescent Surface Biomarkers as Senotherapeutic Targets

**DOI:** 10.3390/cells10071740

**Published:** 2021-07-09

**Authors:** Martina Rossi, Kotb Abdelmohsen

**Affiliations:** Laboratory of Genetics and Genomics, RNA Regulation Section, National Institute on Aging Intramural Research Program, National Institutes of Health, 251 Bayview Blvd., Baltimore, MD 21224, USA; martina.rossi@nih.gov

**Keywords:** senescence, surface proteins, surfaceome, senolytics, senolysis, senostatic, senescent cell clearance, senotherapeutics, senotherapy

## Abstract

Senescence is linked to a wide range of age-associated diseases and physiological declines. Thus, senotherapeutics are emerging to suppress the detrimental effects of senescence either by senomorphics or senolytics. Senomorphics suppress the traits associated with senescence phenotypes, while senolytics aim to clear senescent cells by suppressing their survival and enhancing the apoptotic pathways. The main goal of these approaches is to suppress the proinflammatory senescence-associated secretory phenotype (SASP) and to promote the immune recognition and elimination of senescent cells. One increasingly attractive approach is the targeting of molecules or proteins specifically present on the surface of senescent cells. These proteins may play roles in the maintenance and survival of senescent cells and hence can be targeted for senolysis. In this review, we summarize the recent knowledge regarding senolysis with a focus on novel surface biomarkers of cellular senescence and discuss their emergence as senotherapeutic targets.

## 1. Introduction

Cellular senescence is a phenotype associated with limited replicative capacity and irreversible growth arrest of primary cells first described by Leonard Hayflick in the early 1960s [1]. Senescent cells are characterized by specific phenotypical features including enlarged and flattened cell morphology, enhanced lysosomal beta-galactosidase activity (senescence-associated-beta-galactosidase, SA-β-Gal), increased expression of cell cycle inhibitors (p21Cip1/Waf1, p16Ink4A, p15Ink4B, and p53), and high metabolic activity including GSK3, AMPK, and mTOR pathways [2,3,4]. In this regard, senescent cells differ from quiescent cells, which instead display a reversible cell cycle arrest and are characterized by a low metabolic status, a decrease in glucose uptake, and a reduction in mRNA translation [5].

Senescent cells arise in culture and in tissues following a variety of damaging insults such as DNA damage, oxidative stress, telomere shortening, mitochondrial dysfunction, and aberrant activation of oncoproteins. Besides the role of damaging agents, cellular senescence may also be induced by physiological stimuli including developmental and repair signals. Therefore, transiently-induced senescence plays a beneficial role in physiological events such as organogenesis, tissue homeostasis, and wound repair. Furthermore, the upregulation of cell cycle inhibitors in senescent cells plays a crucial role in their ability to suppress the development of cancer, thus senescence is considered as a tumor-suppressor phenotype [6,7,8].

Alongside the beneficial roles associated with transiently-induced senescence, the accumulation of senescent cells exerts detrimental effects on the functionality of tissues and organs. Indeed, the active metabolism of senescent cells drives the production of several members of cytokines, chemokines, growth factors, and proteases collectively known as senescence-associated-secretory-phenotype (SASP). Members of SASP include pro-inflammatory cytokines and chemokines such as interleukin (IL)-6, IL-8, and matrix degrading enzymes like metalloproteinases (MMP)-1, MMP-3, and MMP-10. The release of such molecules in the cellular microenvironment induces a pro-inflammatory milieu, therefore leading to immune cell recruitment with the reinforcement of inflammation, paracrine senescence, tissue remodeling, and tissue degeneration [9,10].

Cellular senescence is one of the processes contributing to macroscopic aging. With aging, senescent cells accumulate in the body tissues and associate with age-related pathologies, which include, among others, neurodegenerative diseases (Alzheimer’s disease, Parkinson’s disease), atherosclerosis, type 2 diabetes, tissue fibrosis, and cancer [7,8,11]. Given the role of senescence in aging and age-associated diseases, there is a growing interest in developing approaches in order to target senescent cells [12,13]. Much of this effort is focused on the development of strategies aimed at the clearance of senescent cells *in vivo* to reduce their accumulation and the subsequent alterations in tissue functionality. Here, we review the main approaches used to identify senescent cells *in vitro* and *in vivo*, with a focus on novel biomarkers of cellular senescence localized on the cell surface. We discuss the roles of surface proteins in SASP production and in the regulation of immune surveillance. Finally, we highlight the main therapeutic approaches that can be adopted to eliminate senescent cells *in vivo* by pharmacological and genetic approaches, with a focus on targeting the senescent surfaceome.

## 2. Markers of Cellular Senescence

Senescent cells can be detected and characterized based on phenotypical features and the expression of several biochemical markers. Given that there is no universal marker, the establishment of the senescence phenotype is normally supported by multiple markers.

### 2.1. Morphological Changes

Senescent cells display a typical enlarged morphology which is thought to be correlated with the expansion of the ER and with the unfolded protein response (UPR) [14]. Indeed, the primary human fibroblast cells commonly used to study senescence *in vitro* lose their fusiform shape and acquire an irregular shape upon senescence [14]. Other morphological changes associated with senescence include the irregular nuclear shape associated with loss of the nuclear envelope component Lamin B1 [15].

### 2.2. Senescence-Associated β-Galactosidase Activity

The senescence-associated β-galactosidase (SA-β-gal) is a lysosomal enzyme encoded by the gene *GLB1*, which accumulates specifically in senescent cells along with the increase of lysosome content [16]. The activity of SA-β-gal can be measured by adding the chromogenic X-Gal substrate to fixed cells at the suboptimal pH 6.0. Upon conversion of X-gal, cells develop a blue stain that can be visualized under a bright field microscope [17]. Quantitative assays for the detection of SA-β-gal have been also developed and are commercially available. These are based on the introduction of a reagent (SPiDER-β-Gal) that releases fluorescence upon selective processing by SA-β-gal [18].

### 2.3. Cell Cycle Regulators

Cellular senescence is characterized by the permanent arrest of cell proliferation. Therefore, analyzing the expression of cell cycle regulators is important to characterize senescent cells and to determine the molecular pathways actively involved in the cycle arrest. The cell cycle regulators mainly used as markers of senescence include p16, p53, and p21, whose expression levels can be assessed by RT-qPCR, immunoblotting, and immunofluorescence. Proliferation assays and specific markers of proliferation (e.g., the nuclear marker of proliferation Ki67) are commonly used to confirm the reduction in cell proliferation associated with senescence. Additionally, flow cytometry can be used to detect the cell cycle distribution and the specific phase in which senescent cells are arrested [19].

### 2.4. SASP Levels

Senescent cells synthesize and secrete a variety of cytokines, chemokines, growth factors and extracellular matrix (ECM)-remodeling enzymes like metalloproteinases (MMPs), which are robustly expressed in different models of senescence [20]. Cytokine arrays, ELISA kits, and Multiplex technologies are used to measure the levels of soluble molecules including the cytokine IL-6 and the chemokine IL-8 (CXCL8) [20]. Other components of SASP include, but are not limited to, pro-inflammatory IL-1α, IL-1β, and tumor necrosis factor alpha (TNF-α) [21]. Furthermore, different groups of extracellular vesicles (EVs) including exosomes, microvesicles, and apoptotic bodies are highly produced by senescent cells. EVs have been shown to enclose a variety of SASP molecules, DNA, microRNAs, and proteins. These molecules in the EVs have been found to modulate the phenotype and gene expression patterns of the recipient cells [22].

### 2.5. DNA Damage and Telomere Shortening

Classic markers of DNA damage response (DDR) include phosphorylation at Ser139 of the histone protein H2AX, forming gamma-H2AX, which plays a role in recruiting and localizing DNA repair molecules [23,24]. Gamma-H2AX frequently colocalizes with another marker of DNA damage repair, p53-binding protein 1 (53BP1), which is phosphorylated at the residue Ser1219 [24]. Senescence is also associated with chromatin condensation and with the formation of heterochromatin foci (known as senescence-associated heterochromatin foci, SAHF), which repress the expression of genes involved in promoting cell proliferation [25]. SAHF can be identified by fluorescent techniques including DAPI staining and immunofluorescence directed to histone variants like macroH2A, H3K9Me2/3, and HP1 proteins [26,27].

### 2.6. Novel RNA- and Protein-Markers of Cellular Senescence

Recent transcriptomic studies [3,28] have revealed a number of novel transcripts (coding and non-coding RNAs) whose expression is specifically upregulated or downregulated in senescent cells. For example, we performed RNA-sequencing analysis on different models of cellular senescence including human diploid fibroblasts (WI-38, IMR-90) and endothelial cells (HUVEC, HAEC). The integration of these data allowed for the identification of novel RNA markers useful to characterize senescent cells in addition to the classic transcripts upregulated in senescence including *CDKN1A* (*p21*) and *CDKN2A* (*p16*). This analysis also determined shared downregulated transcripts including *HIST1H1,*
*HIST1H1D, HIST1H1E, MCUB*, *PTMA*, *FAM129A*, and *ITPRIPL1*. Shared upregulated transcripts include the lncRNA *PURPL*, and several mRNAs such as *SRPX*, *STAT1*, *CCND3*, *CLDN1*, and *NCSTN* [28]. Although some transcripts like *SRPX* mRNA, have been described in senescence and cancer, the majority of the RNAs in the transcriptome signature remain to be studied in senescence. Similarly, proteomics analysis performed in different types of senescence allowed for the identification of novel protein markers of cellular senescence such as DPP4 and SCAMP4 [29]. While DPP4 promotes cellular senescence, SCAMP4 was found to play a role in SASP secretion, as discussed below.

## 3. Molecular Pathways in Cellular Senescence

The transcription factor TP53 is a tumor suppressor and well-established marker of senescence. It activates the transcription of several cell cycle regulators that prominently include p21 upregulation. In addition, the anti-proliferative program regulated by pRB and p16 inhibits CDKs and transcription factors in the E2F family. Additionally, the mechanistic target of rapamycin (mTOR) pathway plays crucial roles in cell proliferation and metabolic programs that control several biological processes including senescence and lifespan. Thus, these pathways (p53/p21, p16/RB, and mTOR) play essential roles in pro-senescence networks and traits as discussed below.

### 3.1. p53/p21 Pathway

The tumor-suppressor p53 is a master regulator of cellular senescence [30,31]. Different types of DNA damage-inducing agents like ionizing radiation (IR), doxorubicin, and etoposide trigger the stabilization and the activation of p53, which mainly acts as a transcription factor involved in apoptosis, DNA repair, and senescence [32]. One of the well-known p53-target genes is *CDKN1A* (*cyclin dependent kinase inhibitor 1A*), which encodes the cell cycle inhibitor p21^Cip1/Waf1^. p21 inhibits the activation of cycline-dependent kinases (CDKs) at either G1/S or G2/M checkpoints, therefore blocking the cell-cycle progression. Importantly, p21 plays a crucial role in the survival of senescent cells by inhibiting apoptosis through the repression of cell cycle progression. Other studies have revealed the ability of p21 to promote senescent cell survival by restraining the caspase signaling [31,33].

### 3.2. p16/RB Pathway

P16^INK4A^, which is encoded by the gene *CDKN2A* (*cyclin dependent kinase inhibitor 2A*), is one of the pivotal markers of cellular senescence. This cell cycle inhibitor interferes with the cyclin D-CDK4/6 activity, thus maintaining the retinoblastoma (RB) protein in its hypophosphorylated state at the G1 phase. When hypophosphorylated, RB binds and inhibits the transcription factor E2F, which is involved in G1-to-S cell cycle progression [34]. Interestingly, the p16/RB pathway seems to be involved in the long-term maintenance of cellular senescence, whereas the p53/p21 pathway seems to play a key role in the initiation phase of senescence, as suggested by a frequent decrease observed in p53 and p21 levels in late senescence [31].

### 3.3. mTOR Pathway

A growing body of evidence suggests that the mTOR signaling pathway plays a key role in senescence and aging. For instance, inhibition of the mTOR complex 1 (mTORC1) with rapamycin increases lifespan in different organism models [35]. Importantly, the activation of mTOR signaling pathways serves in distinguishing senescent cells from quiescent cells, in which the mTOR pathway is usually not active. The mTOR pathway regulates protein *synthesis* and mitochondrial metabolism. Thus, it plays a central role in senescence due to the fact that senescent cells maintain a highly active metabolism even though they are in an indefinite state of growth arrest [35]. Moreover, the global rate of protein translation in senescent cells is lower compared to proliferating cells and is centered on the production of SASP members including pro-inflammatory cytokines [36]. Importantly, mTOR was shown to also promote senescence via autophagy inhibition, thus resulting in the accumulation of damaged proteins and organelles [35].

## 4. Challenges and Limitations Associated with the Identification of Senescent Cells

Despite the abundance of molecular markers, the identification of senescent cells raises some challenges. First, the current markers are not universally present in all senescent cells and are not entirely specific for the senescence status. For instance, SA-β-gal is active in cells with high lysosomal activity in the absence of senescence and is strongly influenced by cell confluency and oxidative stress [37]. Similarly, the increase in p21 expression occurs in a variety of conditions, which may be independent of senescence such as acute DNA damage due to p53 activation. Indeed, high p21 and p53 levels are believed to be important for the onset of senescence but their elevation is not always maintained after the establishment of the senescence program [19,31]. Finally, some pro-inflammatory cytokines commonly associated with SASP such as IL-6 and IL-8 are upregulated in a variety of inflammatory conditions and are not necessarily associated with senescence [38]. Like other markers, the expression and the secretion of SASP factors is highly variable between different cell types and different senescence programs, and their expression is highly dynamic and highly variable over time [19,39]. Therefore, when studying the senescent phenotype, it is advisable to evaluate multiple markers along with the arrest in cell proliferation.

### 4.1. Identification of Senescence in Tissues Ex Vivo

The identification of senescent cell *ex vivo* raises further challenges compared to the assessment of senescence *in vitro*. First, the well-established senescence marker SA-β-gal requires fresh tissues and cryopreservation, thus limiting the analysis of paraffin-embedded sections due to the lack of enzymatic activity [40,41]. However, there are alternative methods available for the detection of senescent cells in archived tissues. For example, the histochemical staining Sudan-Black-B (SBB) is used to identify lipofuscin, an aggregate of oxidized proteins, lipids, and metals known to accumulate in aged tissues [19,40]. A further challenge for the identification of senescent cells *ex vivo* is due to the fact that the expression of cell-cycle inhibitors like p21 and p16 is highly variable among different tissues and often the expression level is not easily detected by common techniques of immunofluorescence. For example, in a previous study [41], we performed a systematic analysis of senescent cell markers in human tissues across a broad spectrum of ages. We analyzed two classical senescence markers, p16 and p21, to identify senescent cells in tissue arrays designed to include different organs. We found an increase with donor age in both p21-positive and p16-positive cells in epidermis, pancreas, and kidney. In other tissues, there was a high variation in the expression of the two markers: p16-positive cells increased with age in brain cortex, liver, spleen, and colon, while p21-expressing cells increased in the dermis. However, there were no detectable levels of p21- or p16-positive cells in skeletal and cardiac muscle. In addition, p21- or p16-positive cells in lung did not show significant age-dependent changes. In summary, different tissues display different patterns of p16 and p21 as a function of age, probably due to different triggers of senescence programs [41].

### 4.2. Identification of Senescent Cells In Vivo

The development of methods to detect senescence *in vivo* has gained interest for the identification and tracking of senescent cells, and, most importantly, for the potential elimination and clearance of senescent cells in the context of age-associated diseases. Changes in cell morphology and proliferation markers are not commonly used to assess senescence *in vivo*. This is mainly due to (i) the lack of evidence for changes in cell size in live tissues, and (ii) the fact that cells in tissues are often in a quiescent or terminally differentiated state [19]. The identification of senescent cells *ex vivo* mainly relies on flow cytometry using animal models and genetic manipulation for live tracking [19]. For example, transgenic or knock-in mouse models have been developed in order to track, analyze, and eliminate senescent cells in the context of aging and age-related diseases. Since p16 is one of the most-well established markers of senescence, these mouse models are engineered to express a reporter like EGFP or luciferase under the control of the p16 promoter, thus allowing for the identification of senescent cells and the live-tracking of these cells in real-time [6,42,43]. Moreover, the reporter gene is often co-expressed with a protein, which may induce cell death upon treatment with specific drugs administered *in vivo*, thus allowing a better understanding of the impact of senescent cells in age-related diseases [6,42,43]. Importantly, Baker and colleagues developed a transgene for the elimination of p16-positive cells upon drug treatment. They showed that the clearance of senescent cells significantly delayed the onset of age-associated disorders in mice [42].

One of the main advantages of flow cytometry aimed at the identification of surface markers is the ability to preserve the membrane integrity and cell viability, thus, allowing for the isolation and the study of senescent cells without damaging the cells. In addition, flow cytometry allows for the identification of multiple markers of senescence simultaneously, leading to a greater level of confidence in the characterization of senescent cells. Furthermore, several methods have been developed to identify SA-β-gal or nuclear markers with flow cytometry, even though in these specific cases, the detection of intracellular markers would compromise cell viability [19]. Finally, flow cytometry may play a crucial role in addressing the heterogeneity of senescent cells in cultures and tissues. For instance, it can be used for the screening and isolation of subpopulations expressing different sets of senescence markers. In the next sections, we address the importance of surface molecules as novel biomarkers of senescence in more detail.

## 5. The Emergence of Surface Proteins as Biomarkers and Therapeutic Targets

The detection of novel biomarkers of senescence on the cell surface has become increasingly attractive for a variety of reasons [44,45]. First, as previously mentioned, senescent cells expressing specific markers on the surface can be easily identified and isolated by flow cytometry without compromising the membrane integrity and cell viability. Second, and most importantly, surface molecules can be used as therapeutic targets for the administration of drugs, with potential applications *in vivo* for the clearance of senescent cells in age-related disorders. Third, senescence markers expressed on the cell surface can be easily used to classify and analyze certain subpopulations of senescent cells, thus allowing for the study of heterogeneity in senescence. Here we describe and discuss well-characterized surface molecules differentially expressed in senescent cells. Furthermore, we discuss some approaches aimed at the elimination of senescent cells based on the senescence surfaceome.

### 5.1. Novel Biomarkers on Senescent Cell Surface

Althubiti and colleagues performed a screening of plasma membrane-associated proteins preferentially expressed on the surface of senescent cells [46]. Specifically, they used inducible p21- or p16-expression systems in the bladder cancer cell line EJ. They validated a number of these proteins as novel markers for the identification of senescent cells in cultures and tissue samples, namely DEP1, NTAL, EBP50, STX4, VAMP3, ARMCX3, B2MG, LANCL1, PLD3, and VPS26A. The study defined a strategy for the rapid detection of senescence cells in culture through cell sorting using a combination of two markers (DEP1 and B2MG). Further analysis of samples obtained from mouse senescent models revealed that some of the novel markers had an expression pattern comparable to or higher than p16, a classical marker of senescence [46].

Recently, we identified other potential markers of cellular senescence based on the proteomics analysis of molecules enriched on the plasma membrane of senescent WI-38 (human diploid fibroblasts) [29]. The main candidate, DPP4, was further validated as a robust marker of senescence by other studies on cellular senescence and aging [29,47,48]. For instance, a crucial role for DPP4 was revealed in vascular senescence [49], wound healing [50], and bone loss [51]. In addition, we also identified another marker enriched on the surface of senescent cells, SCAMP4, which we found to promote the secretion of SASP in human fibroblasts [52].

Sagiv and colleagues identified two ligands, MICA/B and ULBP2 (NKG2D ligands), consistently upregulated in different models of senescence. These ligands are implicated in immune surveillance by NK cells expressing the NKG2D receptor [53]. Importantly, it has been reported that senescent cells may evade immunosurveillance by suppressing these NKG2D ligands [54]. The implications are not only in the context of cellular senescence, but also in aging and chemotherapy-resistant cancers, in which senescence actively persists [54,55].

Several other surface molecules were validated as markers of senescence by other studies including TNFRSF10D/CD264 [56], NOTCH1 [57], NOTCH3 [58], CD36 [59], oxidized Vimentin [60], ICAM-1 [12], and uPAR [61]. More recently, it was proposed that a broad range of surface markers can be used to identify immunosenescent T-cells in chronic kidney disease [62]. Finally, immunomodulatory signals and surface markers on mesenchymal stem cells (MSCs) were shown to be differentially expressed between old and young mice [63,64]. Here, we briefly describe the function of novel surface markers of senescence identified by these studies (Table 1).

#### 5.1.1. DPP4

DPP4 (Dipeptidyl-peptidase 4) is a glycoprotein ubiquitously expressed on the surface of senescent cells and involved in the cleavage of several substrates including cytokines and growth factors. Due to its role in the regulation of incretins in glucose homeostasis, DPP4 has been well studied in the treatments of diabetes [69,70]. DPP4 has been identified and validated as a robust marker of cellular senescence expressed in different models of senescence [29,48]. DPP4 is transcriptionally induced in oncogene-induced senescence (OIS) by lncRNA-OIS1 [47,71]. The use of DPP4 inhibitors was shown to play a protective role in the vascular system and kidney of aging mice [72,73] and was shown to reduce the senescent phenotype in endotheliocytes [49]. As discussed below, DPP4 seems to play a role in senescent cell survival and can be targeted to eliminate senescent cells [29].

#### 5.1.2. SCAMP4

SCAMP4 (Secretory carrier membrane protein 4) is a protein involved in membrane trafficking [52]. Like DPP4, SCAMP4 is highly expressed on the surface of senescent human diploid fibroblasts. While DPP4 is transcriptionally upregulated, SCAMP4 protein is highly stable in senescent cells compared to proliferating cells. Importantly, silencing SCAMP4 decreased the levels of pro-inflammatory cytokines, while overexpression enhanced the production of SASP components including IL-6 and IL-8 [52]. These findings suggest that SCAMP4 can be targeted to reduce the pro-inflammatory SASP factors in senescence.

#### 5.1.3. DEP1

The *dep1* gene encodes a member of the tyrosine phosphatase family named protein tyrosine phosphatase receptor type J (PTPRJ/CD148). PTPRJ/CD148 has been mainly characterized in leukocytes where it plays a role in cell activation and migration under inflammatory conditions [74,75]. PTPRJ/CD148 activation has been shown to negatively regulate growth factor signals and cell proliferation in different cell types including epithelial cells, endothelial cells, and podocytes [76]. The combined use of DEP1 and beta-2-microglobulin (B2MG) has been shown to be efficient for the rapid detection of senescent cells by flow cytometry [46].

#### 5.1.4. B2MG

Beta-2-microglobulin (B2MG or B2M) is expressed on the surface of almost all nucleated cells and is involved in the presentation of peptide antigens to the immune system. Althubiti and colleagues highlighted the importance of B2MG as a novel marker of senescence [46]. More recently, serum proteomics analysis in an elderly population led to the identification of B2MG as a possible marker of osteoporosis [65]. Importantly, some approaches aimed at clearing senescent cells *in vivo* currently target B2MG-expressing cells through anti-B2MG monoclonal antibodies conjugated with cytotoxic nanoparticles [66].

#### 5.1.5. CD264

The surface molecule CD264 (also known as TNFRSF10D or TRAILR4) acts as a decoy receptor for TRAIL (TNF-related apoptosis-inducing ligand) and has been shown to prevent TRAIL-mediated apoptosis in different cell types. Madsen and colleagues identified CD264 as a surface marker of senescent bone marrow mesenchymal stem cells (hBM-MSCs). Interestingly, CD264 showed an inverse correlation with the regenerative potential of hBM-MSCs [56].

#### 5.1.6. CD36

CD36 is a surface glycoprotein that binds a variety of ligands including lipoproteins, phospholipids, collagen, and thrombospondin. It may act as a scavenger receptor with a role in inflammation, lipid utilization, and fatty acid metabolism. Multiple studies identified CD36 as highly expressed on the surface of senescent cells and proposed a role for CD36 in the SASP production and in the regulation of lipid metabolism in senescent cells [59,67].

#### 5.1.7. ICAM-1

ICAM-1 (Intercellular adhesion molecule-1) is a glycoprotein mainly expressed on endothelial cells and cells of the immune system. ICAM-1 expression increases upon cytokine stimulation and mediates the interaction between endothelial cells and activated leucocytes. ICAM-1 was found to be upregulated by p53 in a variety of senescent cells including human fibroblasts and vascular smooth muscle cells. Importantly, ICAM-1 was also identified in tissue specimens from atherosclerotic lesions [68].

#### 5.1.8. MDA-VIMENTIN

Vimentin is a constituent of intermediate filaments mainly found in mesenchymal cells. The oxidized form of vimentin (malondialdehyde-modified vimentin) was identified on the surface of senescent fibroblasts and in the plasma of age-accelerated mice. Based on these findings, it is suggested that the modified vimentin may play a role in the recognition of senescent cells by the immune system [60].

#### 5.1.9. NOTCH1 and NOTCH3

NOTCH1 and NOTCH3 expression is increased on the surface of senescent cells [57,58]. These receptors are both part of NOTCH signaling, a conserved cell–cell communication pathway involved in embryonic development and malignant transformation [57]. NOTCH1 was found to mediate a switch between two different senescence-associated secretome profiles by regulating the spatial and temporal production of SASP during different phases of senescence induction. Specifically, NOTCH1 was found to increase TGF-b secretion while inhibiting the classic pro-inflammatory cytokines associated with SASP [57]. On the other hand, NOTCH3 was found to promote p21 expression in senescent cells, and its downregulation delayed the onset of senescence in different human cell lines. Conversely, NOTCH3 overexpression was found to induce the senescence phenotype [58].

#### 5.1.10. MICA/B and ULBP2

MICA/B (major histocompatibility complex A/B) and ULBP2 (UL16 binding protein 2) are transmembrane proteins that function as ligands for the NKG2D receptor on NK cells. Healthy normal cells do not constitutively express these ligands; instead, their expression has been shown to be elevated on the surface of malignant, infected, stressed, and senescent cells [53]. The NKG2D receptor plays a role in the detection and elimination of stressed and senescent cells by activating NK-mediated cytotoxicity. Importantly, it has been reported that senescent cells may evade immune surveillance partly by shedding or suppressing their NKG2D ligands [54].

#### 5.1.11. uPAR

The urokinase receptor, also known as urokinase plasminogen activator (uPAR), coordinates intracellular signaling in response to extracellular components. uPAR expression is strongly activated during inflammation, immune responses, stress, and wound healing. Recently, uPAR has been used as a target for the elimination of senescent cells through a novel approach involving T cells engineered to specifically target the urokinase receptor (CAR-T cells) [61]. In the next sections below, we discuss this approach and other strategies targeting surface molecules in more detail.

## 6. Implications of the Surfaceome in Senescence

### 6.1. Senescence Surfaceome and Immune Surveillance

The expression of surface molecules has been shown to play a major role in the recognition and elimination of senescent cells by the immune system. As above-mentioned, senescent cells express, on their surface, ligands for the NKG2D receptor of NK cells including MICA/B and ULBP2, which may drive the detection and elimination of senescent cells by NK-mediated cytotoxicity [53,54]. However, it is known that senescent cells are not always efficiently recognized and cleared by immune cells, therefore, they progressively accumulate in aging tissues [54]. Antibodies directed at MICA or ULBP2 as well as the ablation of the *nkg2d* gene in mice suppressed the elimination of senescent cells by NK cells [53]. Furthermore, it was shown that senescent cells may evade immunosurveillance by suppressing or shedding NKG2D ligands [54]. Indeed, the extracellular domain of several integral membrane proteins can be released in a soluble form by a group of enzymes known as “sheddases”. Two groups of metalloproteases (MMPs and ADAMs) have been shown to play a crucial role in the shedding process of NKG2D ligands. Interestingly, shedding induced by ADAM10 was shown to enhance a senescent phenotype in multiple myeloma, whereas the use of a metalloproteinase inhibitor (Maristat) enhanced NK cell-mediated immune surveillance [55]. Recently, it was shown that senescent cells express a non-canonical MHC (major histocompatibility complex) molecule (HLA-E, HLA class I histocompatibility antigen, alpha chain E). HLA-E interacts with the inhibitory receptor NKG2A expressed by NK and CD8+ T cells and inhibits the immune response against senescent cells [77].

### 6.2. Senescence Surfaceome and SASP 

Being localized at the border between intracellular and extracellular compartments, surface molecules are often involved in the production or regulation of SASP secretion. As above-mentioned, several surface proteins upregulated in senescence including SCAMP4, CD36, and NOTCH1 play roles in the regulation of the SASP profile. For instance, SCAMP4 was found to promote the secretion of classic pro-inflammatory cytokines and chemokines commonly identified in senescence such as IL-6, IL-8, GDF-15, IL-1a, and IL-1b [52]. The scavenger receptor CD36 was reported to interact with several ligands including amyloid beta 42 (Aβ42) and oxidized low-density lipoprotein (oxLDL) to activate the transcription of NF-κB-driven SASP factors [59]. NOTCH1 was shown to mediate a switch between two different secretion profiles, specifically between a pro-inflammatory SASP and a TGF-b rich secretome [57]. An earlier report showed that the interleukin IL-1a may accumulate on the surface of senescent cells (membrane-bound IL-1a) and triggers the production of other cytokines [78]. The use of neutralizing antibodies or receptor antagonists against the membrane-bound IL-1a decreased the levels of secreted IL-6 and IL-8. These findings not only define IL-1 as an upstream regulator of the senescence-associated IL-6/IL-8 cytokine network, but also reinforce the importance of surface molecules in SASP regulation [78].

## 7. Clearance of Senescent Cells

### 7.1. Pharmacological and Genetic Approaches

Up-to-date studies aimed at the pharmacological elimination of senescent cells mainly rely on two classes of compounds called senolytics and senomorphics. Senolytic compounds selectively induce senescent cell death with an ideally minimum effect on proliferating and healthy cells. Senomorphic compounds block the secretory phenotype of senescent cells, therefore reducing the inflammation associated with senescence [79,80]. Senolytic compounds used *in vitro* and *in vivo* include, among others, inhibitors of the BCL-2 family (such as the agents ABT-263 and ABT-737) and a cocktail of two drugs, dasatinib and quercetin (D + Q). Dasatinib is a FDA-approved tyrosine kinase inhibitor, while quercetin is a flavonoid present in many fruits and vegetables [79]. Senomorphic compounds proven to reduced SASP include naturally-occurring flavonoids such as apigenin and kaempferol [80].

Recent studies have highlighted the beneficial effect of senolytic compounds *in vivo* in mouse models. In particular, the senolytic compounds reduce the symptoms associated with atherosclerosis, osteoporosis, and fibrosis, hence resulting in an extended health span [81,82,83]. Importantly, the results of the first-in-human, open-label clinical trial of senolytics recently provided evidence that senolytics are effective in decreasing senescent cells in patients with idiopathic pulmonary fibrosis [84]. Preliminary results from other clinical trials have proven the efficiency of D + Q in alleviating the tissue dysfunction associated with diabetes and chronic kidney disease [85].

Despite the progress made in the field of senotherapeutics, the treatment with senolytics raises some challenges and the research is still in its infancy. Research is now focusing on novel compounds targeting molecules preferentially expressed in senescent cells, in order to minimize the effect on healthy cells [79]. Furthermore, additional insight is necessary to understand the potential long-term side effects of senolytic drugs. One of the limitations is that senolytic and senomorphic compounds show a high degree of variability in their efficiency in different cell lines and tissues, thus making it difficult to adopt one specific senolytic as a widely used strategy to eliminate senescent cells *in vivo* [86].

As above-mentioned, genetic approaches used to identify and target senescent cells *in vivo* mainly rely on transgenic or knockout mouse models. Reporters like EGFP or luciferase are expressed under the control of the p16 promoter to identify and trace senescent cells. The reporter gene is co-expressed with a protein triggering cell death upon treatment with specific drugs allowing specific targeting of senescent cells [6,42,43]. Baker and colleagues [42] developed the novel transgene INK-ATTAC for the elimination of p16-positive cells upon treatment with the drug AP20187. Results showed that the clearance of senescent cells delayed the progression of age-related phenotypes and improved the functionality of skeletal muscle and adipose tissue in old mice [42]. Later in 2014, Demaria and colleagues generated a mouse model (p16-3MR, trimodality reporter) to track senescent cells by luminescence and to eliminate them by ganciclovir administration [6]. Importantly, wound-associated senescent cells were shown to promote optimal wound healing by secreting PDGF-A, a SASP factor that promotes myofibroblast differentiation. Furthermore, elimination of senescent cells delayed cutaneous wound healing, therefore showing a beneficial role for SASP and senescence in tissue repair [6]. These and other studies raise a clear debate aiming at the strategies of senescent cell clearance. Kill or not to kill senescent cells remain to be further investigated for a better understanding of the risks and the benefits.

### 7.2. Targeting the Senescent Surfaceome

One of the main advantages of targeting molecules highly or specifically expressed on the surface of senescent cells is the selectivity of this approach, minimizing the side effects on healthy cells. Furthermore, surface molecules are easily accessible from the bloodstream or the extracellular environment, therefore making it possible to target senescent cells with different therapeutic approaches, as discussed below (Figure 1).

#### 7.2.1. ADCC

With the identification of the membrane protein DPP4 as a biomarker of cellular senescence, Kim and colleagues adopted the antibody-dependent cell-mediated cytotoxicity (ADCC) to selectively eliminate senescent cells *in vitro* without affecting dividing cells. This approach relies on innate immune cells providing cytotoxicity activated by antibodies linked to target cells. Importantly, a high efficiency was observed when using a humanized antibody directed at DPP4, guiding NK cells to selectively kill the antibody-labeled senescent cells [29].

#### 7.2.2. CAR-T Cells

A recent approach adopted to eliminate senescent cells through the surfaceome involves chimeric antigen receptor (CAR) T cells [61]. In this approach, T cells are taken from a patient’s blood and specifically engineered to express an artificial T-cell receptor for a certain protein. CAR-T cells are then grown *ex vivo* and given back to the patient by infusion. Importantly, CAR-T cells that target uPAR, a cell-surface protein upregulated in senescence, have shown to improve the survival and the tissue functionality of mice with liver fibrosis [61].

#### 7.2.3. Neutralizing Antibodies/Nanoparticles

Other potential therapies for the clearance of senescent cells through the surfaceome include neutralizing antibodies and nanoparticles [44]. With this approach, the antibody recognizes and blocks a surface protein, thereby neutralizing its functions. These antibodies can be linked to NK cells as explained above as well as to nanoparticles that are capable of inducing cell death. Recently, senescent cells have been targeted *in vivo* with the use of monoclonal antibodies directed at B2MG and conjugated with engineered nanoparticles [66]. When the antibody recognizes senescent cells, changes in the assembly of the nanoparticles expose granzyme B and induce apoptosis in senescent cells. Elimination of senescent cells with this approach has been shown to improve the tissue homeostasis and the renal function in a mouse model of senescence [66]. The above-mentioned approaches are summarized in Figure 1.

## 8. Closing Remarks and Future Perspectives

Senescent cells accumulate in the human body during the aging process or after exposure to damaging insults like ionizing radiations and chemotherapeutic drugs. Importantly, cellular senescence is considered one of the microscopic processes underlying macroscopic aging and has been shown to exert detrimental effects on tissue and organs. The senescence-associated secretory phenotype, known as SASP, induces a chronic proinflammatory state that leads to persistent immune activation, disruption of the tissue architecture, and tissue degeneration [9,10].

The identification of novel biomarkers of cellular senescence has become increasingly important and attractive due to the potential medicinal applications to combat age-related diseases. Indeed, elimination of senescent cells *in vivo* has proven to be beneficial to delay the onset and the complications of age-related diseases [42,48,84,87].

Several approaches targeting surface molecules are emerging as novel strategies for the elimination of senescent cells. This is mainly due to the fact that surface molecules are selectively expressed on senescent cells, allowing for a high degree of specificity in the context of therapeutic strategies and limiting the side effects on healthy cells. Interestingly, some of these approaches such as CAR-T cells or ADCC have been first developed for the treatments and elimination of cancers [88]. They have been recently utilized to target senescent cells, not only *in vitro* but also *in vivo*. Despite the progresses in this field, research on the senescent surfaceome is still in its infancy. Future studies will acquire more in-depth analyses to validate novel biomarkers that are selectively expressed on the surface of senescent cells in order to minimize the impacts on healthy cells. Another aspect limiting the standardization and the outcome of senescent cell elimination is the cellular heterogeneity [44]. For instance, individual senescent cell populations present different expression profiles and not all cells may possess a high level of a specific molecule on their surface. Future studies will be directed at the characterization of senescent cell subpopulations based also on the expression of surface biomarkers. Surface molecules could therefore be used to define senescent subpopulations, which may respond to different treatments. Furthermore, molecules expressed on the surface of senescent cells may be easily used as antigens to sort different populations by flow cytometry, with the advantage of maintaining cell viability for further studies on cell subpopulations. The choice of surface antigens for cell sorting should take into consideration the presence of large extracellular epitopes, which are easily recognized by commercially available antibodies [46]. A similar concept may apply to the choice of surface molecules used as a target for neutralizing antibodies and antibody-conjugated drugs. Optimal surface molecules used as targets should have well-defined extracellular epitopes with senescent-cell specificity. Due to the lack of markers expressed exclusively on senescent cells, future approaches could use a combination of two or more surface molecules for the selective targeting of senescent cells. Finally, studies on novel surface biomarkers of senescence have focused on protein molecules, but could in principle be extended to other plasma membrane-associated molecules such as lipids and carbohydrates.

In summary, surface molecules represent an attractive target for senolysis and senotherapy, mainly due to their high specificity and accessibility from the external environment. However, further studies are necessary to identify novel biomarkers of senescence, which are selectively expressed in senescent cells, easily targetable with the current and novel interventions, and ideally expressed by different senescent cell populations and subpopulations.

## Figures and Tables

**Figure 1 cells-10-01740-f001:**
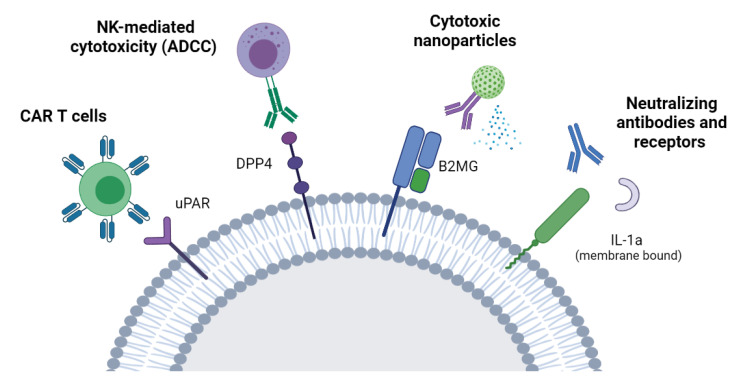
**Targeting senescent cells through the senescent surfaceome.** A schematic of strategies aimed at the elimination of senescent cells by targeting molecules preferentially expressed on the surface of senescent cells. These approaches include NK-mediated cytotoxicity, CAR-T cells, antibody-conjugated nanoparticles, and neutralizing antibodies or soluble receptors. NK: Natural killer cells. ADCC: Antibody-dependent cellular cytotoxicity. CAR-T cells: Chimeric antigen receptor T cells.

**Table 1 cells-10-01740-t001:** Surface molecules as senescence biomarkers implicated in senolysis.

Surface Protein	General Function	Implication in Senescence/Aging	Potential Implication in Senotherapy	Refs.
**DEP1** **PTPRJ** **CD148**	Negative regulation of growth factor signals and cell proliferation.	Biomarkers of senescence.		[46]
**B2MG** **B2M**	Presentation of peptide antigens to the immune system.	Biomarkers of senescence. High levels in serum of elderly population	Target for cytotoxic nanoparticles directed at senescent cells.	[46,65,66]
**CD264** **TNFRSF10D** **TRAILR4**	Antiapoptotic receptor, decoy receptor for TRAIL.	Markers of senescent hBM-MSCs.		[56]
**CD36**	Scavenger receptor with a role in inflammation and lipid metabolism.	Regulation of lipid metabolism.	SASP regulation.	[59,67]
**ICAM-1**	Glycoprotein which mediates the adhesion between endothelial cells and activated leukocytes.	Marker of senescence Increased expression in atherosclerotic lesions	Oxidized form of vimentin, an intermediate filament.	[68]
**MDA-VIMENTIN**	Oxidized form of vimentin, an intermediate filament	Marker of senescence. Increased expression in plasma of age-accelerated mice		[60]
**DPP4** **CD26**	Cleavage of several substrates including cytokines and growth factors. Regulation of incretins in glucose homeostasis.	Biomarker of senescence Protective role on the vascular system and kidney of aging mice	Target for ADCC (NK-mediated cytotoxicity) for the clearance of senescent cells.	[29,47,48]
**NOTCH1**	Member of the NOTCH signaling pathway.	Regulation of different SASP profiles.	SASP regulation.	[57]
**NOTCH3**	Member of the NOTCH signaling pathway.	Regulation of the onset of cellular senescence.		[58]
**SCAMP4**	Secretory protein involved in membrane trafficking.	Regulation of pro-inflammatory SASP.	SASP regulation.	[52]
**MICA/B** **ULBP2**	Ligands for the NKG2D receptor.	Regulation of immune surveillance.	Clearance of senescent cells through NK-mediated cytotoxicity.	[53,54,55]
**uPAR**	Regulation of intracellular signaling in response to extracellular components.	Upregulated in senescence.	Target for the elimination of senescent cells through CAR T cells.	[61]

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
