# Peer review of "The Emergence of Senescent Surface Biomarkers as Senotherapeutic Targets"

_cells, 2021, doi:10.3390/cells10071740_

Round 1
Reviewer 1 Report
Rossi and Abdelmohsen have written a comprehensive informative review on senescence cells focusing on the surface biomarkers of cellular senescence and the potential to use these surface markers as senotherapeutic targets. The Table and Figure are both informative and helpful. The review is concise and clear and provides a novel and unique perspective on this new avenue of research in the field.
I have only minor comments/edits listed below.
Cells formatting has the title hyphenated (Senothera-peutic) I would have this word on one line.
Pg. 3 extracellular vesicles does not need to be capitalized “Furthermore, different groups of Extracellular Vesicles (EVs) are highly produced by senescent cells. For instance, exosomes, microvesicles, and apoptotic bodies, have been shown to enclose a variety of SASP molecules, DNA, microRNAs, and proteins.” I would reword to … “Furthermore, extracellular vesicles (EVs), which is a general term that includes exosomes, microvesicles, and apoptotic bodies, are highly produced by senescent cells. EVs….
2.5 DNA damage response (DDR) doesn’t need to be capitalized.
3.1 this reference didn’t format correctly: (Sammons et al., 2020)
3.1 Cyclin Dependent Kinase Inhibitor 1A doesn’t need to be capitalized
3.2 CDKN2A (Cyclin Dependent Kinase Inhibitor 2A) gene name should be italicized and the name in lower case. Retinoblatoma in lowercase too Also, later in the text in DEP1 section Protein Tyrosine Phosphatase Receptor Type J
DEP1 section leukocytes is spelled wrong
MDA-VIMENTIN section the last sentence probably missing “role”
NOTCH1 and NOTCH3 this first sentence contains some unformatted references
7.1 should be BCL-2
Figure 1 legend “A Schematic…” schematic does not need to be capitalized
Section 8 first sentence is missing…in “the” human body
Just curious is there any evidence that the cell surface proteins are abnormally displayed on the surface (ie. “inside out”)?
Author Response
Rossi and Abdelmohsen have written a comprehensive informative review on senescence cells focusing on the surface biomarkers of cellular senescence and the potential to use these surface markers as senotherapeutic targets. The Table and Figure are both informative and helpful. The review is concise and clear and provides a novel and unique perspective on this new avenue of research in the field. AU. We thank the reviewer for the positive comments.
I have only minor comments/edits listed below.
Cells formatting has the title hyphenated (Senothera-peutic) I would have this word on one line.
AU. We thank the reviewer for this point. The term Senotherapeutic should be all in one line.
Pg. 3 extracellular vesicles does not need to be capitalized “Furthermore, different groups of Extracellular Vesicles (EVs) are highly produced by senescent cells. For instance, exosomes, microvesicles, and apoptotic bodies, have been shown to enclose a variety of SASP molecules, DNA, microRNAs, and proteins.” I would reword to … “Furthermore, extracellular vesicles (EVs), which is a general term that includes exosomes, microvesicles, and apoptotic bodies, are highly produced by senescent cells. EVs….
AU. We thank the reviewer for this comment. The text is revised according to the reviewer's comment, section 2.4.
2.5 DNA damage response (DDR) doesn’t need to be capitalized.
AU. We thank the reviewer for this comment. The text is revised according to the reviewer's comment, section 2.5.
3.1 this reference didn’t format correctly: (Sammons et al., 2020)
AU. We thank the reviewer for this comment. The text is revised according to the reviewer's comment, section 3.1.
3.1 Cyclin Dependent Kinase Inhibitor 1A doesn’t need to be capitalized
AU. We thank the reviewer for this comment. The text is revised according to the reviewer's comment, section 3.1.
3.2 CDKN2A (Cyclin Dependent Kinase Inhibitor 2A) gene name should be italicized and the name in lower case. Retinoblatoma in lowercase too Also, later in the text in DEP1 section Protein Tyrosine Phosphatase Receptor Type J
AU. We thank the reviewer for this comment. The text is revised according to the reviewer's comment, section 3.2.
DEP1 section leukocytes is spelled wrong
AU. We thank the reviewer and apologize for this mistake. The text is revised accordingly, section 5.1, DEP1.
MDA-VIMENTIN section the last sentence probably missing “role”
AU. We thank the reviewer and apologize for this mistake. The text is revised accordingly, section 5.1, MDA-VIMENTIN.
NOTCH1 and NOTCH3 this first sentence contains some unformatted references
AU. We thank the reviewer and apologize for this mistake. The text is revised accordingly, section 5.1, NOTCH1 and NOTCH3.
7.1 should be BCL-2
AU. We thank the reviewer and apologize for this mistake. The text is revised accordingly, section 7.1.
Figure 1 legend “A Schematic…” schematic does not need to be capitalized
AU. We thank the reviewer and apologize for this mistake. The text is revised accordingly.
Section 8 first sentence is missing…in “the” human body
AU. We thank the reviewer and apologize for this mistake. The text is revised accordingly, section 8.
Just curious is there any evidence that the cell surface proteins are abnormally displayed on the surface (ie. “inside out”)?
The reviewer raised an interesting question. Surface proteins like receptors have extracellular domains that bind ligands in order to get activated and transmit signals in the cells. In some cases receptors can be activated intrinsically. However, we are not aware of surface proteins that display an inversed docking on the surface (inside out). Future studies may reveal such situation which could be associated with pathological conditions.

Reviewer 2 Report
The manuscript presents data on senescence surface biomarkers which can be helpful in seeking new targets for drugs used in senotherapy. The manuscript is well written, contains a lot of valuable information, which is presented in a clear and pragmatic manner. It summarizes the currently available markers of senescence and discusses their limitations. Especially, the surface markers are discussed in detail, which is a very useful knowledge from the senotherapeutic point of view. The content of the manuscript is original, collects the existing data in an interesting way, and will surely be of interest to a wide range of readers. The manuscript is valuable especially because, in addition to literature review, the Authors rely on their own research, which greatly increases the quality and reliability of the discussion. The manuscript is prepared very carefully and all the necessary aspects were provided and discussed in a logical and orderly manner. In my opinion, this is a very interesting paper, which meets the requirements for publication in Cells and can be published in its current form.
Author Response
We thank the reviewer for the positive comments.